# Ocean Clutter Characterization Based on PolSAR Data and Second-Order Statistics of Elementary Scatterers

**Georgia Koukiou** * and **Vassilis Anastassopoulos** *

Electronics Laboratory, Physics Department, University of Patras, 26504 Patras, Greece
* Correspondence: gkoukiou@upatras.gr (G.K.); vassilis@upatras.gr (V.A.); Tel.: +30-2610996147 (G.K. & V.A.)

**Abstract:** New features are proposed for sea clutter characterization when PolSAR data are employed. Cameron coherent target decomposition is applied to characterize each SAR pixel by means of the 8 basic elementary scatterers obtained by this decomposition. Since the examined SAR pixel does not match exactly to a specific ideal elementary scatterer, the closest scatterer is assigned to this pixel along with its distance (closeness) from the ideal one. The features proposed are (a) the percentage of each elementary scatterer in the investigated region; (b) the distribution of the closeness of each declared elementary scattering mechanism to its ideal counterpart; (c) the co-occurrence of the same scattering mechanism, taking into consideration its distance from the ideal one; and (d) The co-occurrence of the different scattering mechanisms in pairs, taking again into consideration their distance from the ideal ones. Simultaneously, the skewness and the kurtosis and their 2D versions of the previously mentioned probabilities are employed to further improve ocean clutter characterization. The above features correspond to the sea state condition in each separate region, i.e., the wave height and the wind speed. A clear correspondence between the proposed features and the sea state conditions is established. Data were available from RADARSAT-2 and ALOS-1 PALSAR systems for four different ocean regions on Earth.

**Keywords:** PolSAR data; Cameron decomposition; ocean clutter; sea state conditions; elementary scatterers; co-occurrence matrices

## 1. Introduction

Radar technology has been used to support remote sensing procedures since the Second World War. With the launching of the first satellites, radar and especially SAR contributed to monitoring and recording the surface of the Earth and, simultaneously, to surveillance and detection of targets of interest. Nowadays, Fully Polarimetric SAR imaging (PolSAR) supports detailed surveillance of the Earth's surface, either land or sea, based mainly on the well-established pixel's scattering matrix decomposition methods.

The statistical characterization of the surface of the sea started more than three decades ago, discriminating radar and SAR echoes according to sea state conditions. The K–pdf [1–5] is a compound statistical model describing the radar echoes from the sea surface when the mean value of the pixels is modulated by a different pdf (gamma pdf) than the one describing each separate pixel (Rayleigh pdf). However, for high-resolution radar, each cell contains a small number of scatterers and, consequently, the Rayleigh pdf is not an appropriate distribution to represent each separate pixel due to the weakness of the Central Limit Theorem for application in the case of small number of scatterers. Accordingly, the Generalized Gamma pdf is used instead of the Rayleigh distribution, and, additionally, the same pdf is employed to model the modulation component, leading in this way to the Generalized Compound Model (GC-pdf) [1–3]. In general, the statistics of each separate pixel deviate from Gaussian behavior (Rayleigh amplitude) in the case that, in the area of the pixel, we have a small number of scatterers or one of them is much stronger than the rest [5,6].

With the advancement of PolSAR imaging, Earth's surface was extensively studied, while numerous mathematical approaches for sea state characterization and target detection were developed. In [7], a method is proposed for multidimensional speckle rejection. The method is based on two new matrix models which arise from applying the stochastic summation approach to PolSAR. Methods were investigated in [8] which use PolSAR image data to measure ocean slopes and wave spectra. Another method to measure ocean wave slope spectra using PolSAR data was developed in [9] without the need for a complex hydrodynamic modulation transform function. In [10], the authors considered hybrid PolSAR data of ocean surface slicks and hypothesized that we can design a system that is able to discriminate between mineral oil, plant oil, and clean sea. The authors in [11] reported the results for two polarimetric methods implemented to evaluate their ship detection capabilities. In [12], an efficient algorithm is presented for retrieving the ocean surface wind vector from C-band Radar Satellite RADARSAT-2 PolSAR measurements based upon the co-polarized geophysical model function. A novel method is proposed in [13] to estimate the seawater surface salinity and temperature from the data collected by PolSAR imagery. In [14], PolSAR data are used to estimate the energy dissipation rate under different wind and sea conditions. A shallow-sea topography detection method was proposed based on fully polarimetric Gaofen-3 SAR data [15].

PolSAR images are employed in [16] for discriminating between oil spills and seawater based on the polarization ratio. This polarization ratio method is based on the difference between the scattering mechanism and the dielectric constant for the oil spills compared to that of seawater. For oil spill monitoring, the satellite revisit time needs to be as short as possible to identify minor spills before they can cause widespread damage [17]. Simultaneously, it is necessary to capture a sufficient amount of information about the surface to clearly distinguish between oil-spilled and oil-free sea regions. The hybrid-polarimetry synthetic aperture radar (SAR) system is exploited in [17] for such capabilities. Hybrid-polarized-mode compact polarization (CP) SAR imagery [18] will soon be available with the launch of the RADARSAT Constellation Mission. A methodology is presented in [18] to retrieve the oil–water mixture ratio at the ocean surface using CP SAR imagery. The advantage of the proposed new SAR system is that CP images will have wider swath and shorter revisit time compared to quad-polarization images, which are presently available from space-borne and air-borne SAR.

Millimeter and centimeter electromagnetic waves employed in radar technology can provide information regarding the types of electromagnetic scatterers on the Earth's surface. In particular, coherent and fully polarimetric acquisitions can give detailed information regarding the Earth's surface composition. Accordingly, PolSAR technology is widely used in land cover classification, security tasks, and surveillance [19–23].

Second-order statistics such as the transition matrices corresponding to Markov chains can give better insight into the correlation properties among the PolSAR pixels. Accordingly, the use of discrete Markov processes [3,24,25] outperforms, in detection and classification tasks, conventional distributions for clutter characterization.

Lee and Pottier provide, in [26], a comprehensive interpretation of the polarimetry-analyzing methods for target decomposition that are mainly employed. In [27], PolSAR data are analyzed so that the polarimetry properties of scatterers can contribute to man-made objects detection. The scattering mechanisms characterizing the properties of PolSAR pixels are automatically classified in [28]. Three different matrices, namely, the polarimetric covariance, the coherence, and the Muller matrix, are investigated for each PolSAR pixel in [29], so that covariance symmetries in the PolSAR images are established. Nonnegative eigenvalues are confirmed by means of simple modification when the decomposition of the covariance matrices is performed [30]. Coherent Target Decomposition (CTD) techniques were developed [31–33] for ship detection and identification making use of Cameron's CTD [34]. This was feasible since it was possible to characterize the dominant scattering mechanisms on the ship deck as well as on the surface of the sea for various sea states. Kouroupis and Anastassopoulos in [35] re-investigated the co-diagonalization of the Sinclair

backscattering matrix in order to overcome issues imposed by Huynen decomposition. In this way, the predominant scattering mechanisms are correctly selected. The work presented in [36] employed elementary scattering mechanisms to characterize the pixels belonging to the surface of a ship. These mechanisms are derived from Cameron CTD when the scattering matrix of PolSAR pixels is analyzed. Additionally, the paper introduces a robust feature to discriminate pixels belonging to a ship which is formed by the transition matrix of the first-order Markov chains [37] derived by the alternation of the elementary scatterers on the surface of the ship. Thus, second-order statistics regarding the appearance of the elementary scatterers are used. The work in [38] presents a CFAR detection approach which is based on the features obtained in [36].

In order to classify each PolSAR pixel from the sea surface into a specific elementary scatterer, Cameron decomposition is employed. The pixels, as well as the region of the Earth's surface, are categorized according to the percentage of the elementary scatterers' appearance, their closeness to the corresponding ideal scatterers, and the co-occurrence of the specific elementary scatterers with similar scatterers or with scatterers of a different kind. The closeness of the declared scattering behavior of each pixel to the corresponding ideal elementary scatterer is presented for the whole sea region as a histogram. This gives the capability for evaluating the first four statistical moments of this histogram as well as the 2D skewness and kurtosis of the above-mentioned co-occurrence matrices. It is proved that the sea state conditions affect the statistics and the second-order statistics of the elementary scatterers. A correspondence between the statistical properties of these histograms and the dominating sea conditions is established.

The SNAP software, which is used in our classification procedure, is ideal for analyzing Earth Observation images. It is provided by ESA as open-source software to support its Toolboxes [39]. The SNAP software is appropriate for geocoding and rectification of the remote-sensed images given that ground control points are available. The elementary scatterers map is created when the PolSAR images are geocoded.

The paper is organized as follows. In Section 2, Cameron CTD is analytically presented. In Section 3, the sea PolSAR data are explained in detail. The proposed features as well as the statistical analysis of the PolSAR ocean data and the experimental results are discussed in Section 4. The conclusions given in Section 5 present the main advantages of the proposed method.

## 2. Cameron Decomposition

In recent decades, target decomposition methods were proposed [26–35] in order to analyze the scattering matrix of a PolSAR pixel. Among them, the Cameron CTD method analyzes the behavior of each PolSAR pixel and classifies it into one of the elementary scattering mechanisms [34] with physical meaning. Cameron's coherent decomposition can resolve two essential properties of the PolSAR pixels, namely, reciprocity and symmetry. Reciprocity is an inherent property of monostatic SAR and implies equal non-diagonal elements of the pixel's backscattering matrix. Furthermore, a reciprocal scatterer is symmetric if it has an axis of symmetry in the plane which is normal on the radar Line of Sight (LOS).

Cameron CTD classifies the elementary scatterers according to the properties of reciprocity and symmetry in six categories, namely: trihedral, dihedral, dipole, cylinder, narrow diplane, and $1/4$-wave devices. Two other types of elementary scatterers, left and right helix, represent non-symmetric scattering mechanisms, making the total number of Cameron CTD elementary scatterers eight.

As exposed analytically in [36], Cameron decomposition transforms the backscattering matrix $S$ to the backscattering vector $\vec{S}$, which is further decomposed into a reciprocal $\hat{S}_{\text{rec}}$ and a non-reciprocal component $\hat{S}_{\text{non rec}}$. Moreover, the reciprocal part is being further

decomposed into a maximum symmetric component $\hat{S}_{\text{sym}}^{\text{max}}$ and a minimum symmetric component $\hat{S}_{\text{sym}}^{\text{min}}$. Accordingly, the backscattering vector $\vec{S}$ is as follows:

$$\vec{S} = g\left\{\cos\theta_{\text{rec}}\left\{\cos\left(\tau_{\text{sym}}\right)\hat{S}_{\text{sym}}^{\text{max}} + \sin\left(\tau_{\text{sym}}\right)\hat{S}_{\text{sym}}^{\text{min}}\right\} + \sin\theta_{\text{rec}}\hat{S}_{\text{non rec}}\right\} \qquad (1)$$

where the tone $\vec{S}$ is replaced with $\hat{S}$ for a normalized vector and $g$ represents the total span of the matrix $S$. The degree that the scatterer deviated from the reciprocal space is determined by $\theta_{\text{rec}}$, while the degree of symmetry of the scatterers is described by $\tau_{\text{sym}}$. Due to the fact that the reciprocity theorem always applies, we consider $\theta_{\text{rec}} = 0$, and Equation (1) becomes

$$\vec{S} = g\left\{\cos\tau_{\text{sym}}\hat{S}_{\text{sym}}^{\text{max}} + \sin\tau_{\text{sym}}\hat{S}_{\text{sym}}^{\text{min}}\right\} \qquad (2)$$

The deviation of $\vec{S}$ from $\vec{S}_{\text{sym}}^{\text{max}}$ is expressed by the symmetry degree of the scatterer. A fully symmetric scatterer is represented by $\vec{S}_{\text{sym}}^{\text{max}}$ when $\tau_{\text{sym}} = 0°$, while the fully asymmetric scatterer corresponds to $\tau_{\text{sym}} = 45°$. The component $\hat{S}_{\text{sym}}^{\text{max}}$, which represents maximum symmetry, can be converted into the normalized complex vector $\hat{\Lambda}(z)$, which contains $z$, i.e., the complex parameter describing the scattering mechanism. The complex vector $\hat{\Lambda}(z)$ is as follows:

$$\hat{\Lambda}(z) = \frac{1}{\sqrt{1 + |z|^2}}\begin{bmatrix} 1 \\ 0 \\ 0 \\ z \end{bmatrix} \qquad (3)$$

Table 1 contains the elementary scattering mechanisms and the corresponding values of $z$. The distance given in the following was introduced by Cameron to determine the underlying scattering mechanisms [40].

$$d\left(z, z_{ref}\right) = \sin^{-1}\left(\min\left[d_-\left(z, z_{ref}\right), d_*\left(z, z_{ref}\right)\right]\right) \qquad (4)$$

where

$$d_-(z, z_{\text{ref}}) = \sqrt{\frac{|z - z_{\text{ref}}|^2}{(1 + |z|)^2(1 + |z_{\text{ref}}|)^2}} \qquad (5)$$

and

$$d_*(z, z_{\text{ref}}) = \sqrt{\frac{|z - z_{\text{ref}}^*|^2 + \left(1 - |z|^2\right)\left(1 - |z_{\text{ref}}^*|^2\right)}{(1 + |z|)^2(1 + |z_{\text{ref}}|)^2}} \qquad (6)$$

to estimate the closeness of the measured scatterer parameter $z$ from the reference parameters $z_{ref}$, depicted in Table 1. The symbol $|\dots|$ stands for the modulus of a complex number, while the superscript * stands for complex conjugation. The scatterer in a specific PolSAR pixel is labelled according to the smallest distance obtained from Equation (4). The distance $d\left(z, z_{ref}\right)$ is employed in Section 4 to formulate the statistical features for sea clutter characterization.

**Table 1.** Complex parameter *z* corresponding to elementary scattering mechanisms.

| Complex Parameter $z$ | Normalized Complex Vector $\hat{\Lambda}(z)$ | Scattering Mechanism |
|:---:|:---:|:---:|
| 1 | $\hat{\Lambda}(1)$ | Trihedral |
| $-1$ | $\hat{\Lambda}(-1)$ | Diplane |
| 0 | $\hat{\Lambda}(0)$ | Dipole |
| $+1/2$ | $\hat{\Lambda}(+1/2)$ | Cylinder |
| $-1/2$ | $\hat{\Lambda}(-1/2)$ | Narrow diplane |
| $\pm j$ | $\hat{\Lambda}(\pm j)$ | $1/4$-wave device |

The knowledge to interpret PolSAR imagery based on Cameron's elementary scatterers is essential for understanding land cover types and their electromagnetic echoes response. Accordingly, areas that are flat like sea or bare land correspond to low-entropy pixel groups, while echoes from high-entropy regions like urban regions are strong. One of the color palettes available in MATLAB (JET color map) was employed to map the elementary scatterers in a PolSAR scene. This mapping, which is perceptually quite distinctive, is shown in Table 2. In [41], other types of land cover decompositions are described.

**Table 2.** Cameron 8 Elemental Scatterers color-coding. JET color map is from MATLAB.

| Symmetric Elementary Scatterer | Class | Cameron Color Representation |
|:---:|:---:|:---:|
| Trihedral | 1 | |
| Diplane | 2 | |
| Dipole | 3 | |
| Cylinder | 4 | |
| Narrow Diplane | 5 | |
| 1/4 Wave Device | 6 | |
| Left Helix | 7 | |
| Right Helix | 8 | |

## 3. Sea PolSAR Data

### 3.1. Data Properties

The high-resolution PolSAR imagery comes from two different sources. The first one is the RADARSAT-2 platform (C-band, 5.6 cm), with the main task of all-weather sea and land monitoring. The employed PolSAR imagery is from the Wide Fine Quad-Pol, Single Look Complex, in SLC products [42]. For every PolSAR pixel, with a size of $13.6 \times 7.6$ m, the four polarimetric echoes are provided in complex numbers, having their I and Q components represented by 16-bit accuracy. The Wide Fine Quad-Pol, Single Look Complex SLC RADARSAT-2 data are acquired at an incidence angle of 30–50°. The SNAP software converts the SLC data into geocoded data which possess all corresponding kinds of polarizations, including their I and Q components. Truth maps needed for training and testing purposes are built by employing the geo-referenced images and their corresponding Google Earth maps. An example of such is the fully polarimetric SAR data containing the Vancouver region, which is depicted in Figure 1. The image in Figure 1a corresponds to the amplitude of the I and Q components of the HV part of the fully polarimetric data, while Figure 1b presents the geocoded SAR image registered using the SNAP software on the Google Earth map.

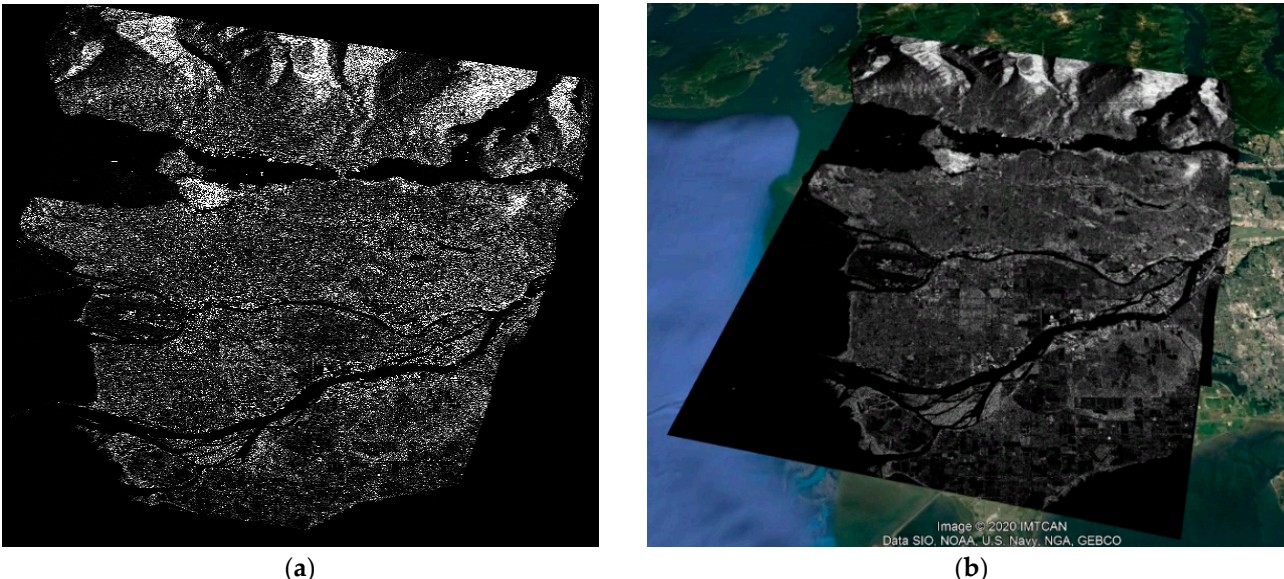

(**a**)　　　　　　　　　　　　　　　　　　　　　(**b**)

**Figure 1.** (**a**) The broader area of Vancouver, British Columbia, Canada. The geocoded data were obtained by means of the SNAP software. The image corresponds to the amplitude of the I and Q components of the HV part of the fully polarimetric data. (**b**) The geocoded SAR image registered using SNAP software on the Google Earth map of the broader area of Vancouver.

The second PolSAR data source was the Phase Array L-band Synthetic Aperture Radar (PALSAR) aboard ALOS-1, using the L-band in the polarimetric mode that provides a nominal resolution of approximately $9.4 \times 3.6$ (m$^2$) (LSC data). ALOS-1 PALSAR acquires data at an incidence angle of $41.5°$. Data from three different ocean regions were acquired so that it was possible to characterize, in each case, the sea state based on the statistical properties of the underlying elementary scatterers. The three regions depicted in Figure 2, are from the central Atlantic, Atlantic equator, and north Atlantic. Sea state conditions can be found from the Copernicus Climate Data Store [43].

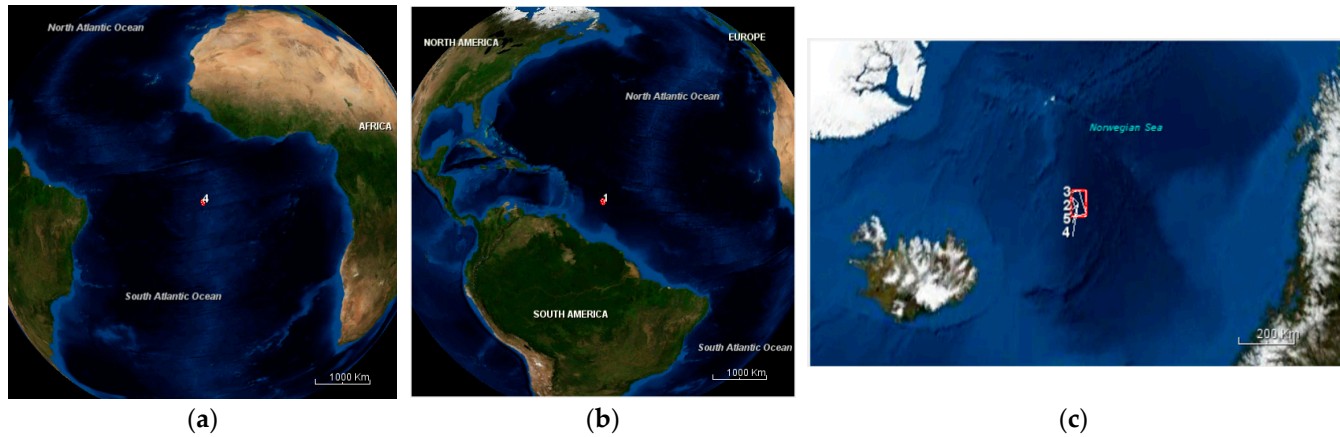

(**a**)　　　　　　　　　　　　(**b**)　　　　　　　　　　　　(**c**)

**Figure 2.** PALSAR ALOS-1 data from the Atlantic Ocean: (**a**) center, (**b**) equator, and (**c**) north.

In Figure 3 that follows statistical measurements are given for the ocean region Atlantic Center. Specifically, the distribution of the closeness (distance) of those SAR pixels characterized as trihedral scatterers from the ideal trihedral is given in Figure 3a. The pixels close to ideal trihedral contribute to the distribution near zero. In Figure 3b is given the co-occurrence matrix for trihedral and cylinder scatterers, with the axes representing the closeness to the corresponding ideal scatterer. Finally, in Figure 3c the same information is provided in a 3D graph.

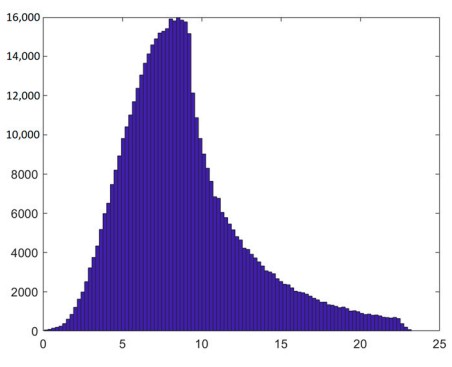

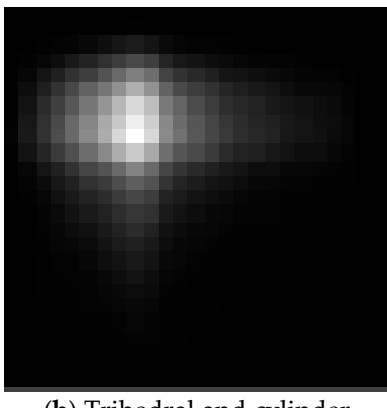

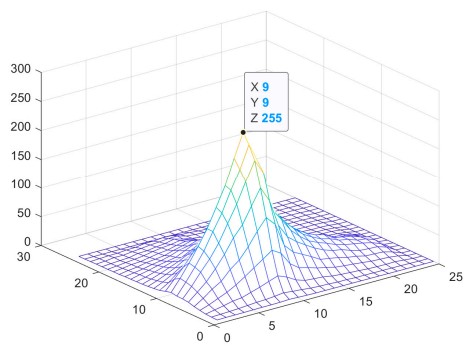

(**a**) Trihedral scatterer 51.2%

(**b**) Trihedral and cylinder scatterers' co-occurrence

(**c**) A 2D graph of the trihedral and cylinder scatterers' co-occurrence

**Figure 3.** Ocean Region: Atlantic Center. (**a**) The distribution of the closeness (distance) of those SAR pixels characterized as trihedral scatterers from the ideal trihedral. The pixels close to ideal trihedral contribute to the distribution near zero. (**b**) Co-occurrence matrix for trihedral and cylinder scatterers, with the axes representing the closeness to the corresponding ideal scatterer. (**c**) The same information is provided in a 3D graph.

### 3.2. SLC Fully Polarimetric Data Preprocessing Stage

SLC fully polarimetric SAR images require special preprocessing [44] in order to convert the data into a form suitable for decomposition and feature extraction for Earth surface classification. The stages followed in this work were the following:

- Radiometric Calibration

Calibration radiometrically corrects a SAR image so that the pixel values truly represent the radar backscatter of the reflecting surface [45]. The corrections that get applied during calibration are mission-specific; therefore, the software automatically determines what the kind of input product is and what corrections need to be applied based on the product's metadata. Calibration is essential for quantitative use of SAR data. During calibration, it is important to save the output product in complex format since Cameron decomposition requires the complex inputs of the Sinclair matrix.

- Polarimetric Matrix Generation

All the polarimetric tools work with either Coherency or Covariance matrices as input. Starting from a Quad Pol SLC product, the Matrix Generation operator converts the product into these types of matrices.

- Polarimetric Speckle Filtering

SAR images are deteriorated by the speckle noise during image capture and transmission. It is a type of multiplicative noise which appears due to the variation in the backscatter from non-homogeneous cells which gives a grainy appearance to SAR images [46]. The backscatter is the result of constructive and destructive interference of the microwave radiation in resolution cells. Many different methods have been discussed in the literature to eliminate speckle noise. The error in the captured image is mainly related to geometry and intensity of pixels. These errors are reformed using definite or statistical mathematical models. The classification of images becomes difficult as the speckle noise manipulates the statistics of the backscatter. It has undesirable effects on the Automatic Recognition System (ATR); hence, speckle filtering has become the first essential step for SAR image processing, such as segmentation and edge detection. Multi-look processing and spatial filtering reduce the speckle noise from images. Multi-look processing is performed at the time of acquisition of low-resolution images. It averages out the speckle noise by taking several "looks" at a target in a single RADAR radius. The average is incoherent. The

single-look processing is quite noisy. Spatial filtering is performed after the acquisition of data.

- Polarimetric Decomposition

In this work, feature extraction utilizes Cameron decomposition. Therefore, each PolSAR cell now corresponds to one elemental scattering mechanism.

- Geometric Correction utilizing Range Doppler Terrain Correction

This is correcting of geometric distortions that lead to geolocation errors. The distortions are induced by side-looking (rather than straight-down-looking or nadir) imaging and are compounded by rugged terrain.

### 3.3. Statistical Feature Extraction

The statistical features used in this work to characterize the distributions of the data used are the mean, the variance, the skewness, and the kurtosis. Second-order skewness and kurtosis are employed to characterize the shape of the co-occurrence matrices due to their 2D nature. For the statistical analysis of the 1D and 2D histograms, the relevant functions of the MATLAB are employed.

In probability theory and statistics [47,48], skewness is a measure of the asymmetry of the probability distribution of a real-valued random variable about its mean. The skewness value can be positive, zero, negative, or undefined. For a unimodal distribution, negative skewness commonly indicates that the tail is on the left side of the distribution, and positive skew indicates that the tail is on the right. In cases where one tail is long but the other tail is fat, skewness does not obey a simple rule. Accordingly:

Negative skew: The left tail is longer. The mass of the distribution is concentrated on the right. The distribution is said to be left-skewed, left-tailed, or skewed to the left, despite the fact that the curve itself appears to be skewed or leaning to the right. A left-skewed distribution usually appears as a right-leaning curve [47].

Positive skew: The right tail is longer. The mass of the distribution is concentrated on the left. The distribution is said to be right-skewed, right-tailed, or skewed to the right, despite the fact that the curve itself appears to be skewed or leaning to the left. A right-skewed distribution usually appears as a left-leaning curve [47].

Skewness in a data series may sometimes be observed not only graphically but by simple inspection of the values. For example, consider the numeric sequence (49, 50, 51), whose values are evenly distributed around a central value of 50. We can transform this sequence into a negatively skewed distribution by adding a value far below the mean, which is probably a negative outlier, e.g., (40, 49, 50, 51). Therefore, the mean of the sequence becomes 47.5, and the median is 49.5. Based on the formula of nonparametric skew, defined as $(\mu - \nu)/\sigma$, the skew is negative. Similarly, we can make the sequence positively skewed by adding a value far above the mean, which is probably a positive outlier, e.g., (49, 50, 51, 60), where the mean is 52.5, and the median is 50.5.

As mentioned earlier, a unimodal distribution with zero value of skewness does not imply that this distribution is symmetric necessarily. However, a symmetric unimodal or multimodal distribution always has zero skewness.

The skewness [47,48] of a random variable $X$ is the third standardized moment $\widetilde{\mu}_3$, defined as

$$\widetilde{\mu} = E\left[\left(\frac{X - \mu}{\sigma}\right)^3\right] = \frac{\mu_3}{\sigma^3} \tag{7}$$

where $\mu$ is the mean, $\sigma$ is the standard deviation, $E$ is the expectation operator, and $\mu_3$ is the third central moment. It is sometimes referred to as Pearson's moment coefficient of skewness, or simply the moment coefficient of skewness, but should not be confused with Pearson's other skewness statistics.

If σ is finite, μ is finite too and skewness can be expressed in terms of the non-central moment $E[X^3]$ by expanding the previous formula:

$$\tilde{\mu}_3 = E\left[\left(\frac{X - \mu}{\sigma}\right)^3\right] = \frac{E[X^3] - 3\mu\sigma^2 - \mu^3}{\sigma^3} \tag{8}$$

Skewness is a descriptive statistic that can be used in conjunction with the histogram and the normal quantile plot to characterize the data or distribution. Skewness indicates the direction and relative magnitude of a distribution's deviation from the normal distribution. With pronounced skewness, standard statistical inference procedures such as a confidence interval for a mean will be not only incorrect, in the sense that the true coverage level will differ from the nominal (e.g., 95%) level, but will also result in unequal error probabilities on each side. Many models assume normal distribution, i.e., the data are symmetric about the mean. The normal distribution has a skewness of zero. But, in reality, data points may not be perfectly symmetric. So, an understanding of the skewness of the dataset indicates whether deviations from the mean are going to be positive or negative. D'Agostino's K-squared test is a goodness-of-fit normality test based on sample skewness and sample kurtosis [48].

In probability theory and statistics, kurtosis is a measure of the "tailedness" of the probability distribution of a real-valued random variable. Like skewness, kurtosis describes a particular aspect of a probability distribution. There are different ways to quantify kurtosis for a theoretical distribution, and there are corresponding ways of estimating it using a sample from a population. Different measures of kurtosis may have different interpretations.

The standard measure of a distribution's kurtosis, originating with Karl Pearson, is a scaled version of the fourth moment of the distribution. This number is related to the tails of the distribution, not its peak. For this measure, higher kurtosis corresponds to a greater extremity of deviations (or outliers) and not the configuration of data near the mean.

The kurtosis is the fourth standardized moment, defined as

$$Kurt[X] = E\left[\left(\frac{X - \mu}{\sigma}\right)^4\right] = \frac{\mu_4}{\sigma^4} \tag{9}$$

where $\mu_4$ is the fourth central moment and $\sigma$ is the standard deviation.

## 4. Statistical Analysis of PolSAR Data and Experimental Results

The statistical analysis of the PolSAR data acquired from the sea surface is carried out based on four different features:

a.  The percentage of the appearance of each dominating elementary scatterer in the specific sea region.
b.  The statistical distribution of the closeness of the estimated dominating elementary scatterer on each separate pixel to the ideal (reference) elementary scatterers. The closeness is evaluated by means of the distance $d\left(z, z_{ref}\right)$ given in Equation (4) and analyzed in Section 2. Accordingly, in a specific region of the ocean, all pixels characterized by the same elementary scatterer contribute to the same distribution describing their distance from this ideal elementary scatterer.
c.  The co-occurrence matrix for encountering the neighboring SAR pixels which correspond to the same elementary scatterer; the matrix axes reveal the closeness to this ideal common elementary scatterer.
d.  The co-occurrence matrix for encountering the neighboring SAR pixels which correspond to different elementary scatterers; the matrix axes reveal the closeness to the corresponding ideal elementary scatterers.

An example is given in Figure 3a regarding the distribution of the closeness (distance) of those SAR pixels characterized as trihedral scatterers from the ideal trihedral. In

Figure 3b is depicted the co-occurrence matrix for trihedral and cylinder scatterers where the axes represent the closeness to the corresponding ideal scatterer. In Figure 3c, the same information as in Figure 3b is provided in a 3D graph.

In the following, experimental measurements are carried out for the three regions in the Atlantic Ocean as well as for the region of Vancouver. Working on the region of Vancouver, measurements were carried out separately for sea and land. The sea near Vancouver is characterized by calm weather conditions compared to those in all three regions of Atlantic. In the following five Figures 4–8, the statistical properties of the five regions are presented regarding the closeness of the more frequently met scatterers to the corresponding ideal ones. Additionally, in the same Figures is given information regarding the location and the time of the data acquisition. In Figure 8, where the statistics of the appearance of the elementary scatterers are shown for the land (mainly) of the Vancouver region, the appearance of the trihedral scatterer is quite small in comparison to that in flat areas like sea. This is obvious from the scatterers' distribution given at the side of the PolSAR HV amplitude image of the region.

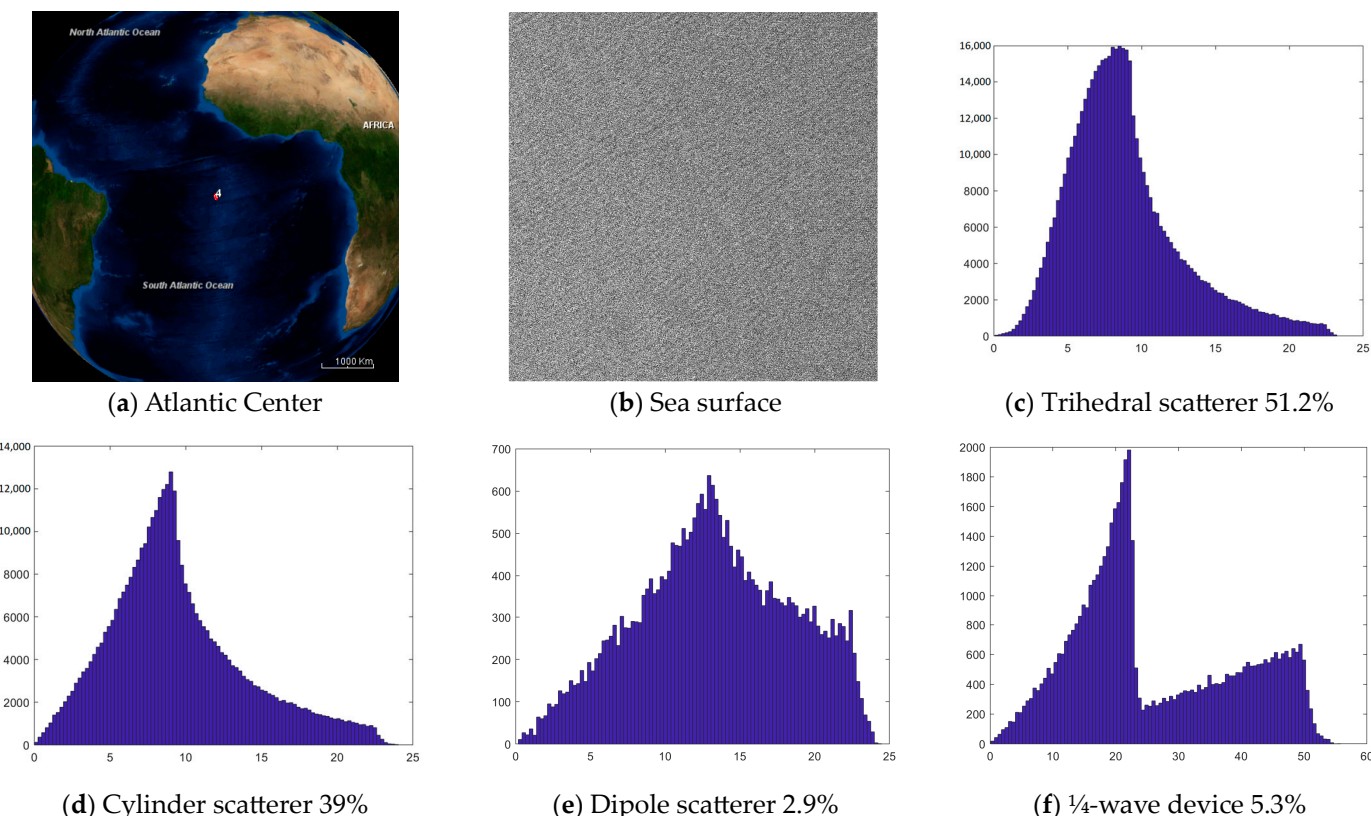

(**a**) Atlantic Center　　　　　　(**b**) Sea surface　　　　　　(**c**) Trihedral scatterer 51.2%

(**d**) Cylinder scatterer 39%　　　(**e**) Dipole scatterer 2.9%　　　(**f**) ¼-wave device 5.3%

**Figure 4.** Region: Atlantic Center, Latitude −5.75, Longitude −14.75, Date-Time 15 January 2011 23:00. (**a**) The location on the Earth's surface, (**b**) image of the sea surface in PolSAR HV amplitude, (**c**) the distribution of the pixels that respond as trihedral elementary scatterers based on the closeness to ideal trihedral, (**d**) the distribution of the pixels that respond as cylinder elementary scatterers based on the closeness to ideal cylinder, (**e**) the distribution of the pixels that respond as dipole elementary scatterers based on the closeness to ideal dipole, and (**f**) the distribution of the pixels that respond as ¼-wave device elementary scatterers based on the closeness to ideal ¼-wave device.

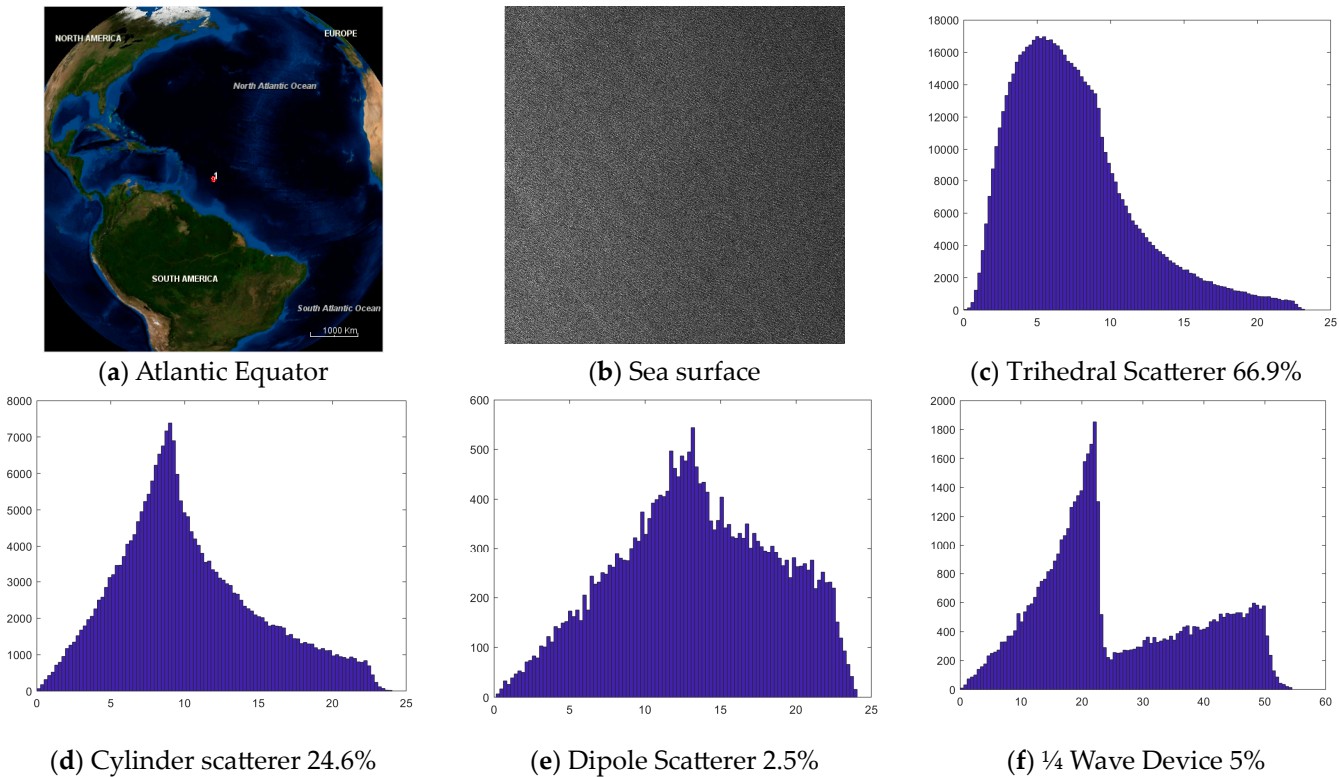

(**a**) Atlantic Equator     (**b**) Sea surface     (**c**) Trihedral Scatterer 66.9%

(**d**) Cylinder scatterer 24.6%     (**e**) Dipole Scatterer 2.5%     (**f**) ¼ Wave Device 5%

**Figure 5.** Region: Atlantic Equator, Latitude −13.25, Longitude −55, Date-Time 2 December 2010 02:00. (**a**) The location on the Earth surface, (**b**) image of the sea surface in PolSAR HV amplitude, (**c**) the distribution of the pixels that respond as trihedral elementary scatterers based on the closeness to ideal trihedral, (**d**) the distribution of the pixels that respond as cylinder elementary scatterers based on the closeness to ideal cylinder, (**e**) the distribution of the pixels that respond as dipole elementary scatterers based on the closeness to ideal dipole, and (**f**) the distribution of the pixels that respond as ¼-wave device elementary scatterers based on the closeness to ideal ¼-wave device.

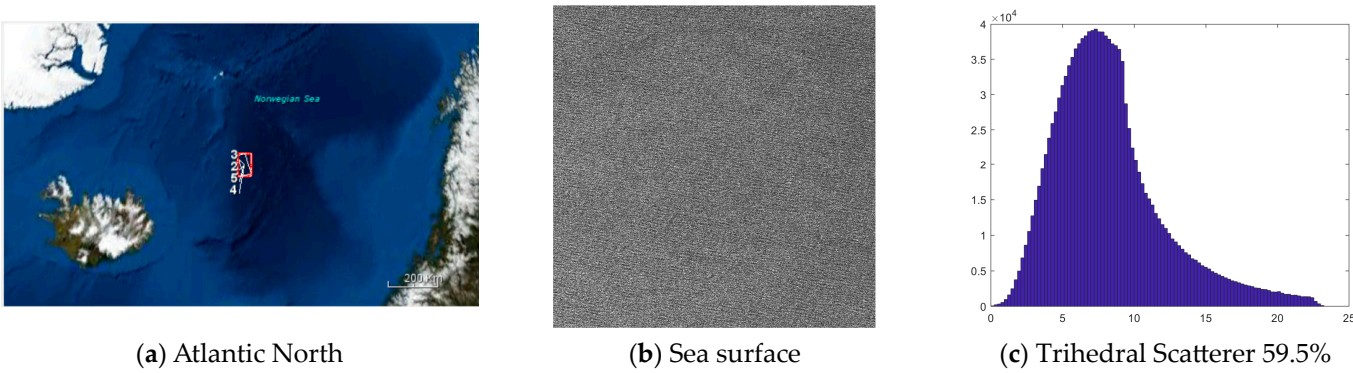

(**a**) Atlantic North     (**b**) Sea surface     (**c**) Trihedral Scatterer 59.5%

**Figure 6.** *Cont.*

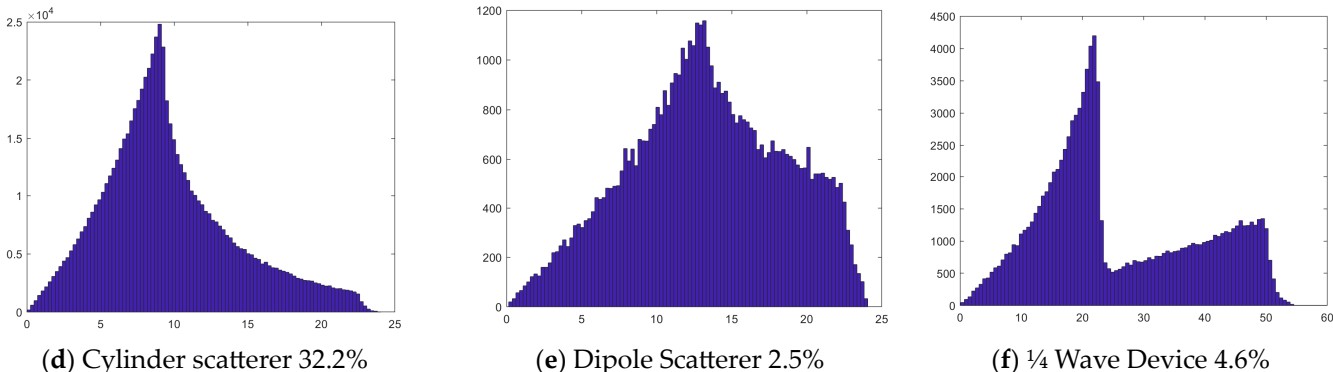

(**d**) Cylinder scatterer 32.2%    (**e**) Dipole Scatterer 2.5%    (**f**) ¼ Wave Device 4.6%

**Figure 6.** Region: Atlantic North, Latitude 68.25, Longitude −6.25, Date-Time 17 November 2009 11:00. (**a**) The location on the Earth surface, (**b**) image of the sea surface in PolSAR HV amplitude, (**c**) the distribution of the pixels that respond as trihedral elementary scatterers based on the closeness to ideal trihedral, (**d**) the distribution of the pixels that respond as cylinder elementary scatterers based on the closeness to ideal cylinder, (**e**) the distribution of the pixels that respond as dipole elementary scatterers based on the closeness to ideal dipole, and (**f**) the distribution of the pixels that respond as ¼-wave device elementary scatterers based on the closeness to ideal ¼-wave device.

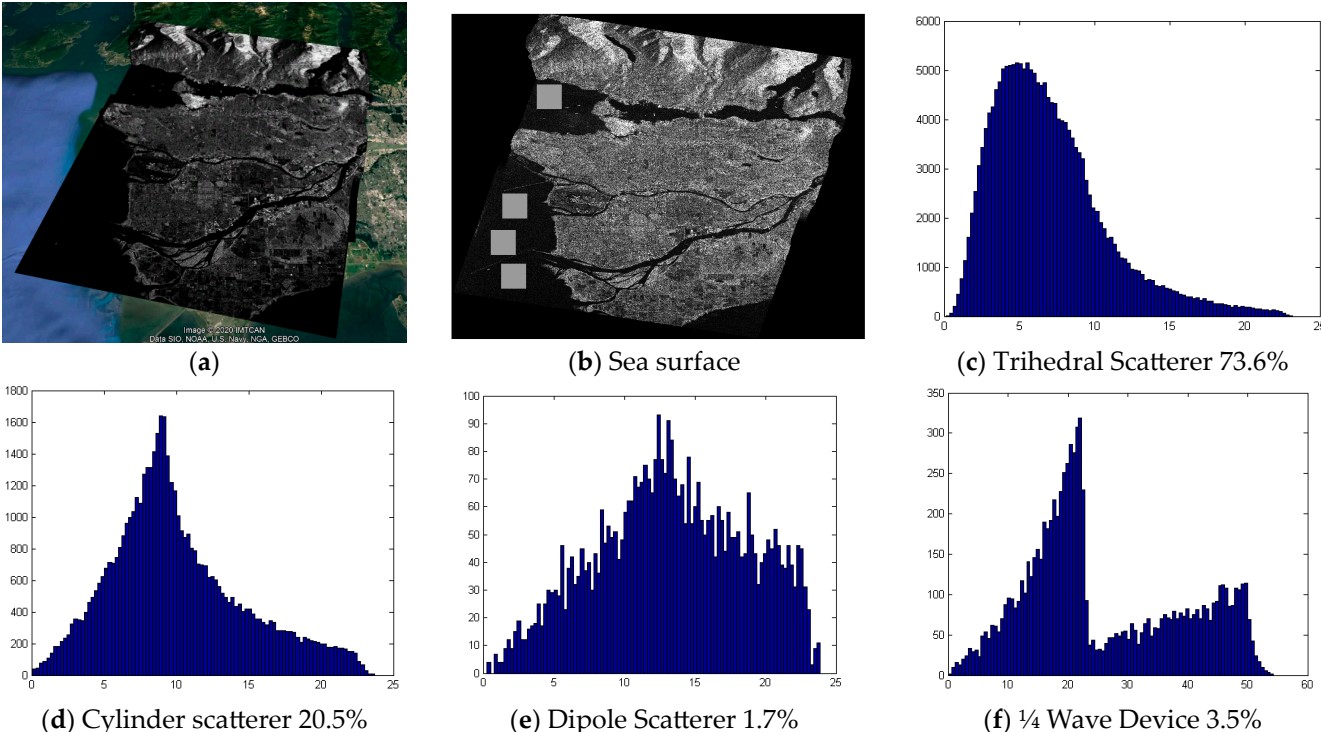

(**d**) Cylinder scatterer 20.5%    (**e**) Dipole Scatterer 1.7%    (**f**) ¼ Wave Device 3.5%

**Figure 7.** Region: Vancouver Sea, Latitude 49, Longitude –123.25, Date-Time 15 April 2008 15:00. (**a**) The location on the Earth surface, (**b**) image of the sea surface in PolSAR HV amplitude (the four squared region in the sea were used), (**c**) the distribution of the pixels that respond as trihedral elementary scatterers based on the closeness to ideal trihedral, (**d**) the distribution of the pixels that respond as cylinder elementary scatterers based on the closeness to ideal cylinder, (**e**) the distribution of the pixels that respond as dipole elementary scatterers based on the closeness to ideal dipole, and (**f**) the distribution of the pixels that respond as ¼-wave device elementary scatterers based on the closeness to ideal ¼-wave device.

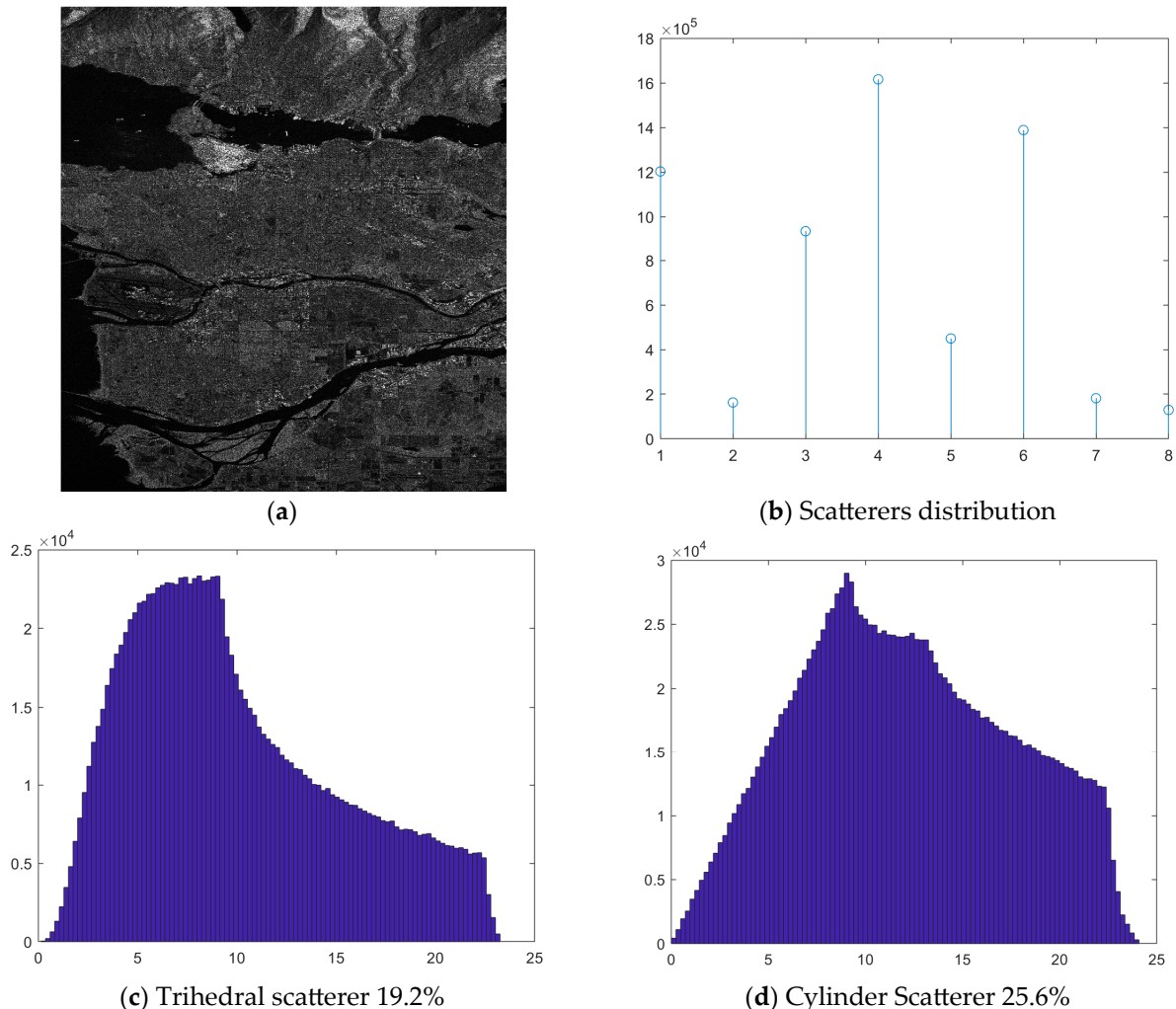

**Figure 8.** Region: Vancouver Land, Latitude 49, Longitude –123.25, Date-Time 15 April 2008 15:00. (**a**) Image of the land surface in PolSAR HV amplitude, (**b**) the distribution of the elementary scatterers in the area of Vancouver, (**c**) the distribution of the pixels that respond as trihedral elementary scatterers based on the closeness to ideal trihedral, (**d**) the distribution of the pixels that respond as cylinder elementary scatterers based on the closeness to ideal cylinder.

According to the material exposed so far and the data available from all sources, Table 3 was constructed. Commenting on the data in Table 3 as well as based on the previous Figures, we can reach the following remarks:

a.   PolSAR pixels at sea are mainly represented by the trihedral elementary scatterer followed by the cylinder. The 1/4-wave device is usually not negligible to around 5%.

b.   For land cover, the percentage of trihedral scatterer is small (<20%) compared to that found on the sea surface (>50%). The percentages of the cylinder, the dipole, the narrow diplane, and the 1/4-wave device are large compared to the percentages of the corresponding elementary scatterers on the sea surface.

c.   In the case of calm sea state conditions, the percentage of the trihedral elementary scatterer becomes quite large (over 70%).

d.   In the open sea, for moderate sea waves height, the percentage of cylinder elementary scatterers increases.

e.   The closeness of each elementary scatterer to the corresponding ideal scatterer, evaluated for each separate PolSAR pixel, does not change significantly with the sea state conditions.

f.   Co-occurrences for the same most frequent elementary scatterer, the trihedral, happens in the calm sea (Vancouver), most frequently between scatterers which are close to the ideal counterparts. Co-occurrences between the most frequent different elementary scatterers (trihedral and cylinder) show no significant difference between calm and rough sea (Figure 9).

**Table 3.** The percentage of the appearance of various elementary scatterers in five different regions along with the Significant Height of combined wind Waves and swell in meters (swh) and the horizontal Wind Speed in m/s (ws).

| Region | Trihedral | Diplane | Dipole | Cylinder | Narrow Diplane | 1/4 Wave Device | Left Helix | Right Helix | swh | ws |
|---|---|---|---|---|---|---|---|---|---|---|
| Atlantic center | 51.2 | 0.2 | 2.9 | 39 | 0.6 | 5.3 | 0.3 | 0.3 | 1.73 | 5 |
| Atlantic Equator | 66.9 | 0.2 | 2.5 | 24.6 | 0.7 | 5 | 0.01 | 0.01 | 2.34 | 6.6 |
| Atlantic North | 59.5 | 0.2 | 2.5 | 32.2 | 0.6 | 4.6 | 0.2 | 0.2 | 2.93 | 6.51 |
| Vancouver Sea | 73.6 | 0.1 | 1.7 | 20.5 | 0.4 | 3.5 | 0.1 | 0.1 | 0.19 | 3.7 |
| Vancouver Sea + Land | 19.2 | 3 | 15.2 | 25.6 | 8 | 23 | 3 | 2.5 | 0.19 | 3.7 |

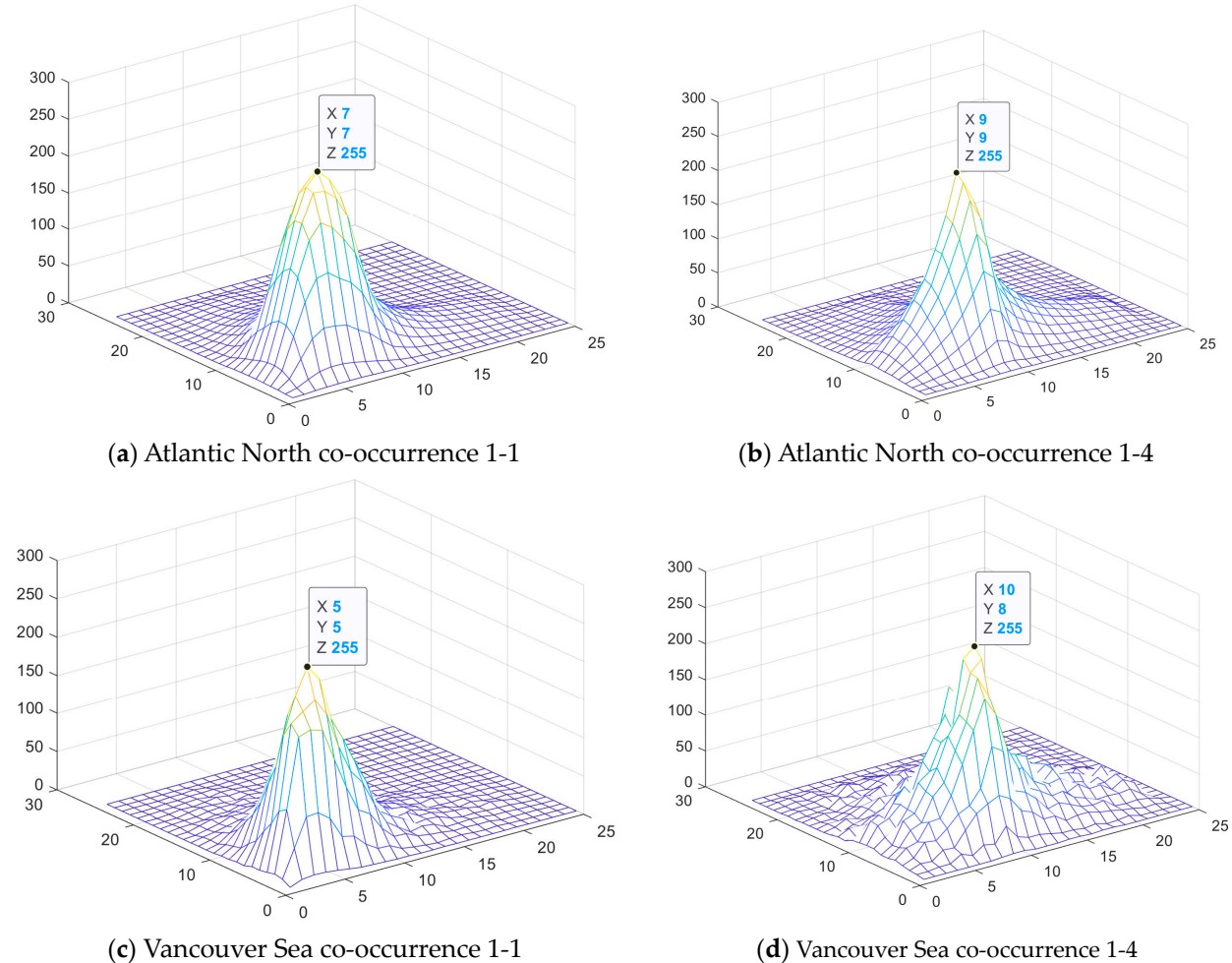

(**a**) Atlantic North co-occurrence 1-1

(**b**) Atlantic North co-occurrence 1-4

(**c**) Vancouver Sea co-occurrence 1-1

(**d**) Vancouver Sea co-occurrence 1-4

**Figure 9.** Co-occurrence matrices for two different sea regions. The upper row comes from Atlantic North while the second row is from the calm Vancouver sea. Co-occurrences for the same most frequent elementary scatterer, the trihedral, happens in the calm sea (Vancouver) most frequently between scatterers which are close to the ideal counterparts (**c**). Co-occurrences between the most frequent different elementary scatterers (trihedral and cylinder) show no significant difference between calm (**d**) and rough sea (**b**).

In order to take advantage of other statistical features for characterizing the sea state of the four ocean regions tested, the four first statistical moments of the distributions

presented in Figures 4–7 were evaluated and are presented in Table 4. These distributions depict the closeness of the declared scatterer in a specific pixel to the corresponding ideal one and represent the most frequent scatterers i.e., the trihedral, the dipole, the cylinder and the 1/4-wave devise. According to what is exposed in Table 4, we mention the following:

a.    The mean for each of the four elementary scatterers does not change significantly from one sea region to another sea region, even if the sea state conditions change.

b.    The variance for the trihedral is large mainly for rough sea state. The variance for the rest of the three elementary scatterers remains unchanged for various sea conditions.

c.    The skewness and the kurtosis slightly increase for the trihedral in calm sea. For the dipole, both the third and fourth moments remain almost unchanged. The same happens for the cylinder and the 1/4-wave device as well.

d.    In general, skewness is of small values (near zero), so the distributions, especially for the dipole, are expected to be symmetric around the mean.

**Table 4.** The four first moments of the distributions presented in Figures 4–7. These distributions depict the closeness of the declared scatterer to the corresponding ideal one and correspond to the most frequent scatterers. The moments are presented for the four different sea regions along with the Significant Height of combined wind Waves and swell in meters (swh) and the horizontal Wind Speed in m/s (ws).

| Region | Moment | Trihedral | Dipole | Cylinder | 1/4 Wave Device | swh | ws |
|---|---|---|---|---|---|---|---|
| Atlantic center | Mean | 8.89 | 13.26 | 9.49 | 25.93 | 1.73 | 5 |
|  | Variance | 15.22 | 26.90 | 20.12 | 169.7 |  |  |
|  | Skewness | 1.02 | −0.08 | 0.72 | 0.43 |  |  |
|  | Kurtosis | 4.10 | 2.33 | 3.30 | 2.0 |  |  |
| Atlantic Equator | Mean | 7.47 | 13.32 | 10.23 | 25.76 | 2.34 | 6.6 |
|  | Variance | 17.11 | 27.33 | 22.47 | 169.02 |  |  |
|  | Skewness | 1.05 | −0.08 | 0.57 | 0.44 |  |  |
|  | Kurtosis | 4.07 | 2.3 | 2.86 | 2.03 |  |  |
| Atlantic North | Mean | 8.26 | 13.3 | 9.66 | 25.85 | 2.93 | 6.51 |
|  | Variance | 14.8 | 26.71 | 20.18 | 168.22 |  |  |
|  | Skewness | 1.08 | −0.09 | 0.7 | 0.44 |  |  |
|  | Kurtosis | 4.36 | 2.34 | 3.25 | 2.03 |  |  |
| Vancouver Sea | Mean | 7.05 | 13.35 | 10.1 | 25.84 | 0.19 | 3.7 |
|  | Variance | 14.83 | 27.99 | 21.35 | 170.09 |  |  |
|  | Skewness | 1.16 | −0.1 | 0.62 | 0.44 |  |  |
|  | Kurtosis | 4.58 | 2.26 | 3.02 | 2.01 |  |  |

The 2D skewness and kurtosis of the co-occurrence matrices presented in Figure 9, for all tested ocean regions, are employed for characterizing the sea clutter in the four ocean regions. The co-occurrence matrices 1-1 and 4-4 correspond to transitions between the same scatterer (trihedral or cylinder) while 1-4 corresponds to transitions between trihedral and cylinder. The results given in Table 5 lead to the following conclusions:

a.    For small wave heights, the skewness and kurtosis for 1-1 co-occurrence is relatively large.

b.    For low wind speed, large values for skewness and kurtosis of the 2D co-occurrences are noted.

c.    The 2D skewness and the kurtosis for the co-occurrence matrices in case of transitions from cylinder to trihedral and vice versa are relatively smaller for low wind speed and wave height.

**Table 5.** Two-dimensional skewness and kurtosis of the co-occurrence matrices presented in Figure 9, for all tested ocean regions. Co-occurrence matrices 1-1 and 4-4 correspond to transitions between the same scatterer (trihedral or cylinder) while 1-4 correspond to transitions between trihedral and cylinder. The Significant Height of combined wind Waves and swell in meters (swh) and the horizontal Wind Speed in m/s (ws) are also given for the corresponding ocean locations.

| Region | Co-Occurrence 1-1 Skewness | Co-Occurrence 1-1 Kurtosis | Co-Occurrence 4-4 Skewness | Co-Occurrence 4-4 Kurtosis | Co-Occurrence 1-4 Skewness | Co-Occurrence 1-4 Kurtosis | swh | ws |
|---|---|---|---|---|---|---|---|---|
| Atlantic center | 3.023 | 17.756 | 2.733 | 11.89 | 2.862 | 12.227 | 1.73 | 5 |
| Atlantic Equator | 2.245 | 7.238 | 2.283 | 9.428 | 2.239 | 8.071 | 2.34 | 6.6 |
| Atlantic North | 3.019 | 11.927 | 2.879 | 13.056 | 3.152 | 14.611 | 2.93 | 6.51 |
| Vancouver Sea | 3.055 | 12.317 | 2.71 | 12.356 | 2.829 | 12.273 | 0.19 | 3.7 |

## 5. Conclusions

The results demonstrated in this work prove that the sea state conditions affect the statistics as well as the second-order statistics of the PolSAR pixels obtained from satellites. Each pixel is represented by an elementary scattering mechanism obtained by Cameron's CTD. A variety of features were employed for this purpose, starting from the simple percentage of the appearance of the elementary scatterers in a sea region, the closeness of each pixel-scatterer to its ideal counterpart, and the second-order statistics (co-occurrence) of each elementary scatterer with similar ones or with scatterers of a specific different kind. Specifically, the four first statistical moments were employed to characterize the distribution which reveals the closeness of each elementary scatterer from the ideal one. Moreover, since co-occurrence matrices of various kinds were evaluated, their 2D skewness and kurtosis were employed to characterize the transitions between various elementary scatterers. The results provided in Tables 3–5 prove that new features are now available for characterizing the sea state conditions. These features can be used as well for ship detection at sea since the statistical behavior of the PolSAR pixels of a ship are totally different from those of the sea surface.

**Author Contributions:** G.K. and V.A. have contributed to this work almost equally in all aspects of its conceptualization, realization, and presentation. All authors have read and agreed to the published version of the manuscript.

**Funding:** This research is partially funded by the Research Committee of the University of Patras.

**Data Availability Statement:** The data are available upon request.

**Conflicts of Interest:** The authors declare no conflict of interest.

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
