# Peer review of "Ocean Clutter Characterization Based on PolSAR Data and Second-Order Statistics of Elementary Scatterers"

_remotesensing, doi:10.3390/rs15112837_

Round 1

Reviewer 1 Report

This paper is all about proposing new features for sea clutter characterization when PolSAR data are employed. The proposed features seems to be novel while features 4b, 4c and 4d appear for the first time in the literature. The proposed features for sea state conditions characterization are based on a well substantiated coherent target decomposition provided by Cameron. The PolSAR data used are from various locations on Earth and provide adequate material for experimentation.

The paper is well written and easy to follow. The abstract and the Introduction seems to be adequate as well as descriptive, and it contains all necessary material to introduce the main idea of the work and simultaneously to address relative works carried out in the past.

Nevertheless, there are some minor weaknesses. To begin, a profound weakness of the paper is that, the authors should make clearer, at the end of section 4, the way that the proposed features help in the classification of different sea state conditions. In particular, the authors have to improve the conclusions giving more details on the experimental results and the importance of the proposed features.

Next, given the fact that the Trihedral and the Cylinder scatterers dominate flat areas as the sea is, I would suggest that the authors should evaluate second or higher order moments (like skewness and Kurtosis) of their distributions which appear in figures 4, 5, 6 and 7 in (c) and (d). Elaboration on these moments could give improved sea clutter classification performance of the proposed features.

Furthermore, 2nd higher-order moments (like 2D skewness and Kurtosis) would also contribute in improved sea clutter classification performance. Working examples would be the 2D curves in Figure 9.

To conclude, the authors have to improve their conclusions by giving more details on the experimental results as well as to highlight the importance of the proposed features.

Reviewer 2 Report

The manuscript lacks novelty in the present form as it just analyses the already known Cameron decompositions approach to study the scattering behaviours of ocean surfaces for different scenarios. The presented work needs significant modifications to make it suitable for this journal. I have listed below my suggestions:

1)  The analysis presented is good, but the end discussion is not conclusive and application-oriented.  It is suggested to add complete descriptions and the proper significance of the obtained results.

2) Explain the possible reason why the appearance of considered (eight different classes of) elementary scatterers are not consistent in the five different cases shown in Table 3 of the paper. 

3) If you suggest that the wind speed and the height of combined wind waves and swell play a role in changing the scattering response, and you are able to find any direct connection between them, it is suggested to add them here.

4) I can see the usefulness of this work, especially in the applications of oil-spill detection presented in the following papers, as the wind speed has a critical role in generating different scattering mechanisms leading to the oil-spill descriptors derivations. (a) doi: 10.3390/rs15071855 (b) doi: 10.1109/LGRS.2023.3258224.

5) The radar incidence angle can also play a vital role in characterizing the scattering response. Therefore, mention the incidence angle information of all the used datasets also, especially in Table 3, for the readers to have a wider view of the results.    

6) Cameron decomposition, which is implemented in this manuscript, is suitable for the full-polarimetry case, as it can be implemented on a 2*2 scattering matrix. However, in recent papers, dual-polarimetry or compact-polarimetry is suggested for ocean surface monitoring because it can provide better monitoring capability. How do you determine whether this work suits this particular category? 

Minor corrections are recommended. For example, I can see "ALOS" in some places and "Alos" in some places for the same satellite sensor name. You can cross-check the whole paper and perform these kinds of corrections.  

Reviewer 3 Report

Both the details of classification process (training/testing ratio, classification method, parameter optimization, accuracy analysis) and the details of SAR data pre-processing (calibration, speckle filtering or orthorectification missing and not presented. Such missing details make the interpretation of the results very difficult and questionable. These details needs to be presented/included in the manuscript. The pre-processing is the most important step before the analysis of the data. 

Round 2

Reviewer 2 Report

The authors have provided a detailed response to the concerns raised by my side. I recommend publishing it in the RS journal. Just a few minor suggestions are mentioned below:

a) In my previous round of revision, I recommended adding the incidence angle of the data acquisition, as it plays an important role in ocean surface characterization.  The authors have added the incidence angle of ALOS-1 data but not for RADARSAT-2 data implemented in the manuscript. Please add this, too, for the convenience of the readers. 

b) Minor corrections in English writing are required. For example, the following sentence must be modified to clarify its meaning.

 ". . .Moreover, since co-occurrence matrices various kinds were evaluated their 2D skewness and kurtosis were employed to characterize the transitions . . ."

Please go through the whole manuscript again and correct it.
